# Patient-Derived Organoids of Colorectal Cancer: A Useful Tool for Personalized Medicine

**DOI:** 10.3390/jpm12050695

**Published:** 2022-04-26

**Authors:** Takumi Kiwaki, Hiroaki Kataoka

**Affiliations:** Section of Oncopathology and Regenerative Biology, Department of Pathology, Faculty of Medicine, University of Miyazaki, 5200 Kihara, Miyazaki 889-1692, Japan; mejina@med.miyazaki-u.ac.jp

**Keywords:** colorectal cancer, patient-derived organoids, 3D culture, personalized medicine

## Abstract

Colorectal cancer is one of the most important malignancies worldwide, with high incidence and mortality rates. Several studies have been conducted using two-dimensional cultured cell lines; however, these cells do not represent a study model of patient tumors very well. In recent years, advancements in three-dimensional culture methods have facilitated the establishment of patient-derived organoids, which have become indispensable for molecular biology-related studies of colorectal cancer. Patient-derived organoids are useful in both basic science and clinical practice; they can help predict the sensitivity of patients with cancer to chemotherapy and radiotherapy and provide the right treatment to the right patient. Regarding precision medicine, combining gene panel testing and organoid-based screening can increase the effectiveness of medical care. In this study, we review the development of three-dimensional culture methods and present the most recent information on the clinical application of patient-derived organoids. Moreover, we discuss the problems and future prospects of organoid-based personalized medicine.

## 1. Introduction

Colorectal cancer (CRC) is one of the most commonly diagnosed cancers globally and the second most common cause of death, causing approximately one million deaths annually [1,2,3]. The number of patients with CRC is expected to increase by 60% by 2030 [3]. Adenocarcinoma is the most common type of CRC, originating from mucosal epithelial cells upon the accumulation of genetic abnormalities over an estimated period of 10–15 years [1]. There are two distinct molecular pathways involved in CRC development; one is the traditional adenoma–carcinoma pathway, which is associated with chromosomal instability and mutations in specific tumor suppressor genes and oncogenes (e.g., *APC*, *RAS*, and *TP53*), and the other is the serrated pathway, which involves the CpG island methylation phenotype and microsatellite instability [4]. Four consensus molecular subtypes (CMSs) have been defined, helping to comprehend biological complexity and heterogeneity of CRC: CMS1, which is hypermutated and strongly immunogenic; CMS2, which shows chromosomal instability and WNT/MYC activation; CMS3, which shows metabolic dysregulation and *KRAS* mutation; and CMS4, which is characterized by CpG hypermethylation, TGF-β activation, stromal invasion, and angiogenesis [5]. In patients with CRC, CMS is correlated with prognosis and response to treatment, which emphasizes the importance of elucidating the molecular biology of CRC for its effective management. Surgical resection, chemoradiotherapy (CRT), targeted therapy, and immunotherapy are the most widely used methods to treat CRC. Although advancements in treatment have prolonged patient survival, the prognosis of patients with metastatic disease remains poor [1]. Several studies have been conducted to better understand the biology of CRC as well as to improve treatment outcomes. One major study topic is the application of three-dimensional (3D) culture models represented by patient-derived organoids (PDOs) [6]. In this article, we review the history of 3D culture systems, their applications in CRC, and the progress of research. In particular, we discuss their relationship with the currently emerging field of personalized medicine.

## 2. In Vitro Models of Colorectal Cancer: From Two to Three Dimensions

Presently, two-dimensional (2D) cultured CRC cell lines are widely used in cancer biology-related investigations [7,8]. They are valuable tools for mechanistic studies of cell growth, analyses of colorectal carcinogenesis, and exploration of anticancer compounds [9]. These cell lines were vigorously established during the 1970s and 1980s, and CRC cell lines such as Caco-2, HT-29, HCT-116, and SW480 are commonly used today [10,11,12]. Generally, primary culture of CRC cells involves isolating fresh tumor tissues via enzymatic treatment and culturing them in fetal bovine serum supplemented media [13]. Obtaining cell lines that can undergo stable passage is challenging; Bian et al. mention that the success rate of cell line establishment from a fresh CRC tumor tissue is only approximately 10% [14]. Each 2D-cultured CRC cell line shows distinct morphological features. As evident from phase-contrast microscopy, cells spread out in sheets (e.g., Caco-2 and HCT-116) or grow as clusters (e.g., NCI-H508) [7]. The molecular features of CRC are well represented by cell lines, with several cell lines showing microsatellite instability and the CpG island methylation phenotype [7]. Furthermore, representative gene alternations of CRC involving *TP53*, *BRAF*, *KRAS*, *PIK3CA*, and *PTEN* have been identified in some cell lines [8]. When 18 frequently used CRC cell lines were classified based on CMS, >3 cell lines were classifiable into all four subtypes (CMS1–4) [15], indicating that a cell line library with sufficient genetic diversity exists to investigate CRCs. Although 2D-cultured CRC cell lines are indispensable tools for cancer research, they do not represent a study model of CRC for several reasons. For instance, they are monoclonal and lack stromal components; moreover, they are morphologically different from actual tumors [7]. CRC shows histological features such as a complicated glandular structure and mucin production, and has a tumor stroma consisting of a mixture of cancer-associated fibroblasts (CAFs), vascular endothelial cells, and immune cells; however, these features are almost impossible to reproduce in 2D-culture. Their cross-contamination is another major problem; there have been cases where CRC cell lines with the same origin have been differently named (e.g., WiDr and HT-29, DLD-1 and HCT-15) [16,17]. Considering that many established cell lines have undergone unspecified long-term passaging in vitro, they do not adequately represent the molecular heterogeneity of patent tumors [18]. Additionally, 2D-cultured cells and actual tumor cells have been found to show different protein expression profiles involving cell proliferation, metabolism, and anticancer drug resistance [19,20,21].

To overcome the issues associated with 2D culture systems and improve understanding of cancer biology, an increasing amount of attention is being paid to 3D culture systems [22,23,24]. Three-dimensional culture systems successfully reproduce the characteristics of patient tumors by maintaining cell–cell interaction and showing oxygen and nutrient gradients similar to those of tissues in vivo [25]. Although many efforts have been made to establish methods for the 3D culture of intestinal epithelium [26,27,28], stable and reproducible long-term growth of intestinal epithelium has proven difficult. In 2009, Sato et al. reported the first long-term organoid culture of intestinal epithelium [29]. They successfully established organoids with crypt–villus structures from Lgr5+ mouse intestinal stem cells, which were then embedded in a certain substrate and provided diverse soluble factors. Sato et al. applied this organoid culture method to human normal intestinal epithelium, mouse colon adenomas, and human CRC [6]. Laminin is enriched in basement membranes surrounding the crypts, and laminin-rich Matrigel was utilized as a substrate [30]; based on previous studies [31,32,33], culture media were supplemented with R-spondin 1, epidermal growth factor, and Noggin. The combination of Matrigel and these three compounds is widely used today for organoid culture of the intestinal epithelium and CRC. Figure 1 shows a PDO established from a patient with CRC; it can be seen that its morphology is similar to that of the original tumor. Intriguingly, in most cases of CRC organoid culture does not need to be supplemented with any growth factors [6], which reflects the high proliferative potential of cancer cells. Several other 3D culture methods have been developed for intestinal epithelium. Ootani et al. reported a robust long-term 3D culture methodology for gastrointestinal culture which incorporated epithelial as well as mesenchymal/stromal components into a collagen-based air-liquid interface 3D culture system [34]. Spheroid cultures for generating cell clusters without any extracellular matrix are used as a 3D culture method as well [35,36,37]. Kondo et al. reported a cancer tissue-originated spheroid-based method; briefly, they prepared cancer organoids by dissociating tumor tissues and allowing them to form spheroids in suspension [38]. Cell-cell contact could be retained during the preparation process, as this offers various advantages. These culture methods can be divided into three categories based on the presence or absence of gel substrate and the method of embedding, namely, Matrigel embedding, air-liquid interfaces, and spheroid culture (Table 1 and Figure 2).

In recent years, organoids have been widely used in various fields of cancer research, which has consequently improved our understanding of driver gene mutations in CRC and oncogenic pathways (e.g., the adenoma-carcinoma sequence, serrated pathway, and inflammation-associated carcinogenesis) [39,40,41,42,43]. Using CRISPR-Cas9-based genome-editing in organoids, CRC carcinogenesis processes of the adenoma-carcinoma sequence pathway and the serrated pathway have been reproduced successfully [39,44]. Additionally, organoids have a wide range of other applications; for example, they have been used to investigate the mechanisms of invasive growth and metastasis [45,46,47,48,49,50,51], crosstalk between cancer cells and stromal cells in shaping the tumor microenvironment [52,53,54], and stem cell-like properties and heterogeneity of cancer cells [55,56].

In addition, organoids can be used for drug screening to identify novel drugs. Drug screening assays based on 2D-cultured cell line panels such as the National Cancer Institute panel of 60 human tumor cell lines and the Cancer Cell Line Encyclopedia have facilitated elucidation of the relationship between drug sensitivity and molecular profiles of cancer cell lines and led to the identification of numerous compounds for novel anticancer drug therapies [57,58]. However, the majority of these compounds fail to progress through clinical trials, and only a few have been commercialized as therapeutic agents, which further emphasizes the inadequacy of 2D-cultured cell lines as a preclinical model [59]. Furthermore, the fact that the expression profile of proteins associated with cancer cell proliferation and drug resistance differs from that of primary tumors greatly impacts the reliability of 2D-cultured cell lines as a preclinical model [19,20]. In contrast, 3D-cultured cell lines can serve as an effective preclinical model, as they can more accurately reproduce the in vivo features of tumors. For example, several studies comparing 2D and 3D culture systems involving the same cells have reported that drug sensitivity differs between the systems and that 3D-cultured cells can better reproduce stem cell-like properties [60,61]. Three-dimensional culture models include patient-derived xenografts, patient tumors transplanted into mice or other animals, and PDOs [62]. Of all the 3D culture systems, PDOs represent the most reliable model, as they recapitulate the in vivo features of tumors at the genomic, transcriptomic, and proteomic levels; moreover, they are easier to handle than animal-based models [15,63,64]. PDO-based drug screening experiments have already been reported, and clinical gene-drug interactions have been identified in anticancer drugs, suggestive of the potential of PDOs as a preclinical model for screening anticancer drugs [65,66,67,68]. Additionally, the establishment of several PDO libraries has been reported, facilitating assessments that completely reflect the phenotypic diversity of CRC cells in a real-would setting [45,69,70]. Although there are no large-scale data that confirm the suitability of PDOs as a preclinical model, drug screening protocols are transitioning from using 2D-cultured cell lines to using PDOs [62], unlike lung cancer, for which molecular targeted therapy has been established for each driver gene, for CRC, the options for molecular targeted therapy remains limited (e.g., EGFR inhibitors, VEGF inhibitors, and immune checkpoint inhibitors) [1,71]. Therefore, it is vital to develop novel therapies using high-throughput drug screening approaches, including drug repositioning, in which PDOs can play a significant role [72].

## 3. Patient-Derived Organoids as Predictive Biomarkers in Cancer Therapy

To optimize medical care for patients with CRC, it is key to identify predictive biomarkers that can help decide which therapy is the most likely to be effective [73]. Commonly assessed predictive biomarkers of cancer therapy include specific protein expression levels (e.g., membranous expression of HER2 is associated with the response to HER2 inhibitor [74,75]), somatic DNA alterations in a single gene (e.g., response to BRAF inhibitors is associated with *BRAF* V600E mutations [76,77]), genome-wide patterns of somatic DNA alterations (e.g., immune checkpoint inhibitor therapy is effective for cancers with high tumor mutation burden [78,79]), and nontumor cell populations that form the tumor microenvironment (e.g., number of tumor-infiltrating lymphocytes is a predictor of the response to immune checkpoint inhibitor [80,81]). Biomarker-matched therapies offer a significant survival benefit in several types of cancers, including CRC [82,83,84]. However, numerous biomarkers are associated with issues such as unreliability of the assay techniques and incomplete of representation of patient clinical outcomes [85]; thus, the identification of robust and reliable new biomarkers is very much needed.

In recent years, PDO-based predicted therapy has been developed for various types of cancers, including CRC. Among the articles published to date, clinical studies that have established PDOs from CRC patients for the correlation study of drug sensitivity of PDO to the corresponding patient’s response to treatment are summarized in Table 2. A PDO-based test was found to predict sensitivity to irinotecan-based chemotherapy for CRC in >80% of patients [86]. Furthermore, it has been suggested that PDOs can help to determine whether patients will benefit from irinotecan therapy; however, it has been reported elsewhere that organoids fail to predict the response of treatment with 5-fluorouracil plus oxaliplatin. Thus, their usefulness seems to vary depending on the type of drugs being used [86]. Wang et al. established PDOs from specimens of patients with stage 4 CRC, determined the sensitivity of PDOs to chemotherapy regimens (i.e., fluorouracil plus oxaliplatin, fluorouracil plus irinotecan, and fluorouracil plus oxaliplatin/capecitabine and oxaliplatin), and compared the response of patients who received the same chemotherapy regimen [87]. Overall, 71 patients were enrolled; PDOs were successfully obtained from 57, with the median time from specimen collection to drug testing being nine days. Their PDO model could accurately predict the response to chemotherapy regimens in 79.69% (51/64) of patients. Susceptibility prediction using PDOs has been investigated for both chemotherapy and radiotherapy. Yao et al. generated organoids from 80 patients with locally advanced rectal cancer who were treated with neoadjuvant CRT [88]. The response to CRT in patients was highly matched to that of PDOs (84.43% accuracy). Moreover, the PDO-based prediction model is expected to be applicable to other solid tumors. For instance, PDOs have been reported to predict the response of patients with esophageal cancer to neoadjuvant CRT [89]. Methods other than PDO have been found to be associated with a relatively low prediction accuracy; for example, predicting the effect of CRT based on imaging findings showed an accuracy of 30–60%, and serum biomarkers showed an accuracy of approximately 70% [90,91]. Conversely, PDOs could predict chemotherapy and radiotherapy response in patients with cancer with a high accuracy of approximately 80%. Consequently, PDO-based prediction seems beneficial in avoiding unnecessary or inappropriate treatment (Figure 3). In cases of CRC, multidrug chemotherapy (e.g., fluorouracil plus oxaliplatin, fluorouracil plus irinotecan, and fluorouracil plus oxaliplatin/capecitabine and oxaliplatin) is widely used as standard treatment; however, its efficacy varies from patient to patient [1]. While preoperative neoadjuvant CRT is used to treat locally advanced rectal cancer, treatment response remains highly variable [92]. Using PDOs to classify CRC patients with diverse biological characteristics thus appears to be an effective method to achieve better responses to chemotherapy and radiotherapy. A high success rate of primary culture is required for consistent clinical application; reported success rates are 63.5–85.7% [86,87,88]. Yao et al. reported that culture failure cases included bacterial contamination and mucinous carcinoma; they failed to culture either of two cases of mucinous carcinoma [88]. On the other hand, Wang et al. concluded that success rates for adenocarcinoma and mucinous carcinoma were 79.54% (58/73) and 82.61% (19/23), respectively, with no significant differences [87]. In addition, Fujii et al. successfully cultured several mucinous carcinoma cases [70]. Therefore, it may be possible to culture mucinous carcinoma; however, care should be taken in specimen collection because the amount and proportion of tumor cells are both lower than those of adenocarcinoma. Other causes of culture failure remain unknown, and require further investigation.

## 4. Patient-Derived Organoids in Precision Medicine

Precision medicine involves matching the right drugs to the right patients; in cancer, there is a tendency to make precision medicine synonymous with genome-based cancer therapeutic matching [94]. Next-generation sequencing (NGS) has enabled the simultaneous examination of hundreds of genes, and a variety of NGS-based tumor-profiling multiplex gene panels have been developed [95,96,97]. Such tests are used in clinical practice, and >50% of patients are detected to show actionable gene alterations; however, only a limited number benefit from therapies recommended by gene panel testing [98,99,100,101]. This dissociation between patients with actionable gene alterations and those who actually receive effective treatment indicates that gene panel testing-based drug selection remains inadequate; thus, alternative screening methods are much needed.

Multiple approaches have been tested to improve the accuracy and effectiveness of precision medicine, one of which is PDO-based drug screening. Following the establishment of organoid-based screening methods involving multidrug panels [67], it has become possible to screen drugs using PDOs and administer the right drugs to the right patients, giving rise to the concept of PDO-based precision medicine (Figure 3). Furthermore, patient samples, i.e., fresh tumor tissues, can be simultaneously prepared for both NGS-based gene panel testing and PDO establishment, and the effectiveness of NGS-suggested drug can be assessed by PDO-based drug screening; consequently, PDO-validated drugs can be administered to patients [102]. Although little has been reported regarding this approach in clinical practice, Ooft et al. conducted a small clinical trial to assess PDO-based drug screening [93]. They enrolled 61 patients with CRC and generated 31 PDOs, of which 25 were subjected to drug screening; 19 PDOs exhibited a response to one or more drugs. Six patients eventually underwent treatment with drugs selected using this PDO-based assay. The treatment response of patients was limited in the trial; in order to accumulate additional data, similar trials with a larger sample size need to be conducted in the future. PDO-based drug screening may overcome the limitations associated with gene alteration-based approaches, resulting in the expansion of treatment options for patients with CRC.

## 5. Challenges and Prospects for the Clinical Application of Patient-Derived Organoid-Based Therapy

We now move to a discussion of the challenges associated with the clinical application of PDO-based personalized therapy and the efforts being made to overcome these challenges. First, attention needs to be given to the handling of clinical samples for PDO establishment. Tumor tissues should be immediately placed in an ice-cold buffer (e.g., phosphate-buffered saline) or culture media. Although it is desirable to begin the culture process soon after specimen collection, this may not always be feasible considering the busy schedule of clinicians and limited access to laboratory facilities. Research has been conducted on the refrigeration or freezing of samples, in an attempt to help prolong the duration between sample collection and culture initiation. For example, Ashley et al. found that refrigerating CRC tissue overnight before culture initiation did not significantly affect the success rate of the culture [103]. In addition, He et al. examined organoid cultures after cryopreservation [104]. They minced resected tissues, which were then placed in a freezing medium and frozen using the slow freezing method for 24 h, followed by storage in liquid nitrogen. The tissues were thawed after more than a year and used for organoid culture; cryopreservation was found to have no significant effect on the success rate of organoid culture. However, it should be noted that the tissues used in their study did not include CRC, and the effects of freezing on CRC tissues remain to be investigated. Although more research is needed in order to standardize storage methods, refrigerating or freezing samples can effectively address the aforementioned limitations. Standardization of culture methods is another challenge that needs to be addressed for clinical application. To be approved through clinical trials and commercially used as a standard method, cultures must be reproducibly and uniformly maintained. The original method reported by Sato and Clevers et al. is widely used for organoid culture today, and a guideline has been published for the culture method [105]. While their method was originally developed for intestinal epithelium culture, the method has long been adapted to the organoid culture of CRC tissue. One concern in this method may be the usage of Matrigel. Matrigel is often used as a substrate in the current organoid culture method; however, it is an animal-derived substrate that is extracted from Engelbreth-Holm-Swarm mouse sarcomas, and standardization of the method may prove difficult considering its complex, poorly defined, and variable composition [106,107,108]. An increasing amount of research is being conducted to identify an alternative substrate to Matrigel; for example, type I collagen has been reported to support organoid formation as well as Matrigel [109]. Moreover, QGel CN99, a novel fully defined hydrogel-based matrix was found to show organoid-forming efficacy comparable with that of Matrigel [110]. Both substrates have been shown to grow intestinal epithelium and could be applied to CRC culture. As little information currently exists on the comparison of these compounds, more studies are warranted to identify a suitable alternative to Matrigel as well as to further standardize culture methods.

Surgical specimens are preferable for PDO culture because of their large tumor tissue volume, although biopsy specimens can be used [45]. In the case of patients with CRC, colon and rectum tissue samples can be easily harvested via biopsy with minimal invasion. Obtaining a biopsy specimen of the primary tumor is feasible even in cases of advanced CRC that cannot be surgically resected, and preparing pretreatment and posttreatment PDOs is possible when a patient undergoes neoadjuvant therapy. PDOs established from metastatic sites harbor driver mutations in the primary lesion which are sufficient for personalized medicine applications [45]. Considering the possibility of additional oncogenic and druggable mutations, using samples from metastatic or recurrent foci in the case of patients with metastasis or recurrence seems preferable. Circulating tumor cells (CTCs), i.e., tumor cells present in the blood of patients with solid tumors, are another source for organoid culture [111,112]. It has been reported that CTCs were detectable in 42% of the patients with metastatic CRC [113]. PDO establishment from CTCs has already been reported in prostate cancer [114]. Although further research is needed, CTC-based organoid culture is minimally invasive, and thus provides considerable benefits to patients.

As manual 3D cultures are highly time consuming, developing a high-throughput and automated method that can stably process many samples is desirable for the clinical application of PDO-based personalized medicine. In this context, Boehnke et al. digested CRC organoids into single cells, suspended them in Matrigel, and plated them into 384-well plates using a robotic arm [115]. Their platform was robust and ensured reproducibility. A culture platform named “hydro-organoid”, where CRC cells self-organize into organoids in a culture medium supplemented with 2% Matrigel, has been used for high-throughput CRC analyses [116]. In comparison with the conventional Matrigel-embedding method these hydro-organoids were able to produce more uniform organoids, enabling reliable drug screening. Additionally, microfluidic platforms and 3D bioprinting have emerged as effective high-throughput methods for CRC organoid establishment and drug screening, and they reportedly show better operability [117,118]. Notably, such high-throughput analyses were conducted using organoids already established using the conventional Matrigel-embedding method; further technological advancements are needed to automate the entire process from specimen collection to drug screening. Drug screening assays generally measure the efficacy of drugs by detecting organoid size or enzymatic activity. Hirashita et al. established a simple method to evaluate the efficacy of an MEK inhibitor which involved immunostaining-based detection of a change in phosphorylation of ribosomal protein S6 [119]. We believe that such a method can improve the reproducibility and simplicity of organoid-based drug screening.

In recent years, cancer immunotherapies such as immune checkpoint inhibition therapy have become a very important topic in cancer treatment and achieved significant clinical success [80,120]. Although conventional molecular targeted therapies target proteins expressed in cancer cells themselves, cancer immunotherapy exerts its cytotoxic effect via immune cells such as T lymphocytes [120,121]. Hence, predicting the efficacy of cancer immunotherapy using conventional organoids consisting only of cancer cells is insufficient; a model that represents the entire tumor immune microenvironment is essential [122,123]. Hong et al. established PDOs that retained autologous tumor-infiltrating lymphoid cells from partially dissociated (diameter >100 µm) CRC tumor tissues [124]. Although the number of lymphoid cells decreased through serial passage, the model was sufficient for the evaluation of immune checkpoint inhibitors, and PDOs derived from a patient with microsatellite instable CRC were sensitive to anti-PD-1 and anti-PD-L1 antibody treatment. Takahashi et al. established a high-throughput system to evaluate immune responses to cancer cells by co-culturing lung cancer organoids with T lymphocytes and natural killer cells [125]. This simple method can be applied for CRC research. Other components of the tumor microenvironment, such as CAFs and tumor-associated macrophages (TAMs), are important for the application of PDOs in personalized medicine. Certain CAFs secrete growth factors such as hepatocyte growth factor and fibroblast growth factor and promote cancer cell proliferation [126]. TAMs contribute to tumor growth by, for example, mediating tumor angiogenesis [127]. Therapeutic strategies (e.g., hepatocyte growth factor inhibitors and angiogenesis inhibitors) targeting these components of the tumor microenvironment have been investigated and will likely become available for clinical applications in the near future [127,128]. Thus, it is vital to establish an assay system that can validate drugs acting via CAFs and TAMs. This is expected to result from the establishing of models that reproduce the tumor microenvironment for the evaluation of cancer immunotherapy and CAF/TAM-targeting therapies. Furthermore, it has recently been reported that gut microbiota are closely related to CRC development and progression [129]; reproducing a microenvironment that consists of both host cells and gut microbiota seems inevitable, and further research is warranted in this regard.

## 6. Conclusions

CRC is one of the most prominent malignant tumors, and is associated with high incidence and mortality rates; its importance is bound to increase with the increasing number of patients across the globe. Over the past decade, advancements in 3D culture methods for intestinal epithelium have provided deep insights into the molecular biology of CRC. Furthermore, 3D cultures, particularly the use of PDOs, are a useful tool that appears promising for widespread clinical application. PDOs can be used as a predictive biomarker to select the right treatment method for the right patient, which can help avoid unnecessary interventions. PDOs may provide additional information in the context of precision medicine, offering new alternatives for patients for whom the standard of care has been exhausted and no treatment is available. Ethical considerations are essential for the utilizations of PDOs, as they contain patients’ genetic information. Although many problems remain to be addressed, these tiny spheres on the culture dish will bring many rewards in the future.

## Figures and Tables

**Figure 1 jpm-12-00695-f001:**
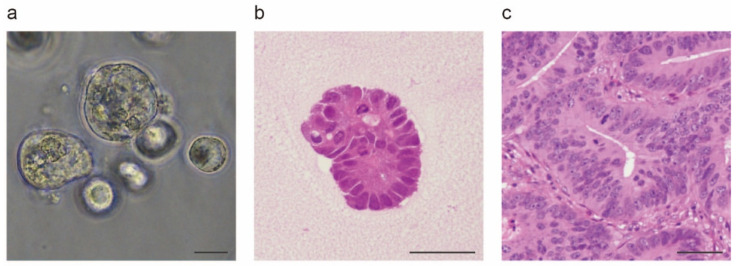
Representative morphology of patient-derived organoids (PDOs). A PDO line, MPL-CR1, was established by the authors from cecal adenocarcinoma with a sample obtained from a 51-year-old male patient. (**a**) Phase-contrast microscopy images showing sphere-like aggregates of cancer cells (scale 100 µm); (**b**) formalin-embedded specimen of the organoid, hematoxylin and eosin (H&E) staining (scale 50 µm); (**c**) Histology of the original tumor, where it can be seen that the morphology of the PDOs is similar to that of the original tumor (H&E staining, scale, 50 µm).

**Figure 2 jpm-12-00695-f002:**
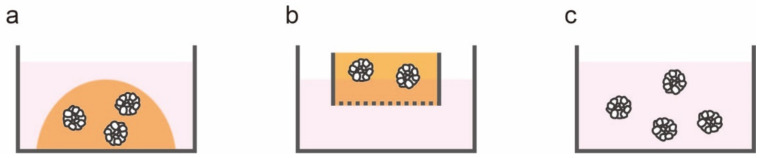
Representative three-dimensional culture methods: (**a**) Matrigel-embedding method; (**b**) collagen-based air-liquid interface system; (**c**) spheroid culture.

**Figure 3 jpm-12-00695-f003:**
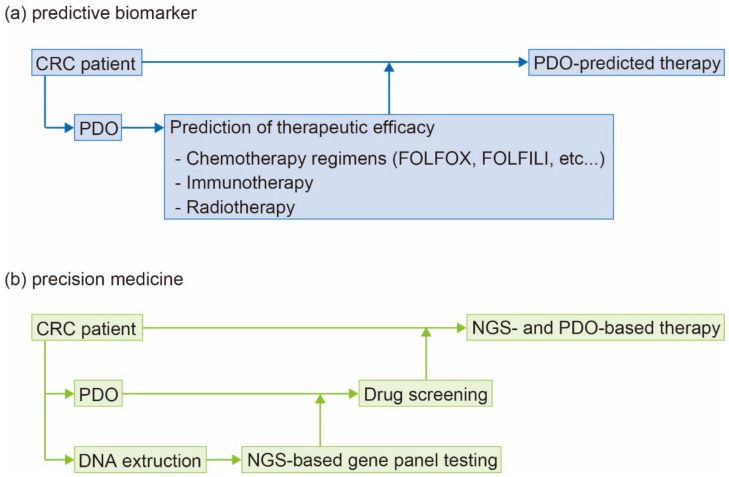
Potential clinical uses of patient-derived organoids. CRC; colorectal cancer, PDO; patient-derived organoid, FOLFOX; fluorouracil plus oxaliplatin, FOLFIRI; fluorouracil plus irinotecan, NGS; next-generation sequencing.

**Table 1 jpm-12-00695-t001:** Brief procedure and characteristics of colorectal cancer (CRC) models.

Method	Procedure	Merit	Demerit
3D culture models			
Matrigel embedding [6]	- Digest samples with trypsin - Embed in Matrigel and seed in a plate	- Used in many studies and well-validated - Genetic manipulation can be performed - Modeling of early-stage CRC is possible	- Matrigel is animal-derived and poorly defined
Air-liquid interface [34]	- Mince samples mechanically - Embed in collagen gel and pour into inner dish - Place inner dish into outer dish containing culture medium	- Ensures an abundant oxygen supply	- Procedure is complicated - Fewer applications in cancer research
Spheroid culture [38]	- Digest samples partially - Floating culture of cell clusters in a noncoated dish	- Partial digestion preserves cell–cell adhesion - Easy to handle, as the procedure is gel-free	- Difficult to reproduce cancer microenvironment
Other models			
2D culture	- Dissolve samples enzymatically - Culture on adhesive plates	- Experimental handling is easy - Genetic modification methods are well-established - The lowest cost of any of these models	- Low similarity to patient tumors - Limited culture success rate
Patient- derived xenografts	- Transplant patient’s tissue into immunodeficient mice	- High similarity to patient tumors - Pre-existing tumor microenvironment - Biologically relevant pharmacokinetics	- Requires long time for establishment - High experimental costs - Concern for animal ethics

**Table 2 jpm-12-00695-t002:** Reported research on personalized medicine for CRC using PDO.

Study	Enrolled Patients	Culture Success Rate	Outcome
Predictive biomarker			
Ooft et al. (2019) [86]	Metastatic CRC	63.5% (40/63)	- PDO predicted efficacy of irinotecan monotherapy with 80% accuracy - PDO predicted efficacy of 5-FU-irinotecan combination therapy with 83.3% accuracy - PDO could not predict response to 5-FU-oxaliplatin combination therapy
Yao et al. (2020) [88]	Locally advanced rectal cancer	85.7% (96/112)	- PDO sensitivity for irinotecan, 5-FU, and irradiation was correlated with clinical outcomes of chemoradiotherapy, with 84.43% accuracy
Wang et al. (2021) [87]	Stage IV CRC	80.2% (77/96)	- PDOs predicted response to chemotherapy with 79.69% accuracy
Precision medicine			
Ooft et al. (2021) [93]	Metastatic CRC	57.4% (31/54)	- Of 25 drug screens performed, 19 showed a response to one or more drugs - Three patients who exhibited organoid response were treated with vistusertib and three were treated with capivasertib, however, they did not demonstrate any objective response

## Data Availability

The data presented in this study are available on request from the corresponding author.

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
