# Peer review of "Patient-Derived Organoids of Colorectal Cancer: A Useful Tool for Personalized Medicine"

_jpm, 2022, doi:10.3390/jpm12050695_

Round 1

Reviewer 1 Report

Comments on the manuscript:
1. In the text and in Table 2 in resume, the authors compare patient-derived colorectal cancer model systems. To focus on the aim of the manuscript I would recommend to the authors expand Table 1 and its discussion in the text for example about the evaluation of drug screening; susceptibility for genetic manipulation; modeling early-stage CRC; etc. in the three model systems.
2. Could the authors explain why they chose to divide typical 3D culture methods in Table 1 on -  Matrigel embedding; Air-liquid interface and Cancer tissue originated spheroids?       
3. Can the authors indicate what are the criteria for the selection of so-called "representative studies" in Table 3?
4. It would be good if in the "Conclusion" part the authors involve a brief comment on the ethical concerns of PDO research.

Author Response

Comment 1: In the text and in Table 2 in resume, the authors compare patient-derived colorectal cancer model systems. To focus on the aim of the manuscript I would recommend to the authors expand Table 1 and its discussion in the text for example about the evaluation of drug screening; susceptibility for genetic manipulation; modeling early-stage CRC; etc. in the three model systems.

Answer: We appreciate the reviewer’s suggestion. According to the suggestion, we have expanded Table 1 and merged with Table 2, making a new Table. We have also added contents regarding genetic modification and early-stage CRC to the table.

Comment 2: Could the authors explain why they chose to divide typical 3D culture methods in Table 1 on -  Matrigel embedding; Air-liquid interface and Cancer tissue originated spheroids?

Answer: We have divided these methods based on the presence or absence of gel substrate and the method of embedding. We have indicated it in the manuscript.

Comment 3: Can the authors indicate what are the criteria for the selection of so-called "representative studies" in Table 3?

Answer: Among the articles published to date, we have chosen clinical studies that have established PDOs from CRC patients for the correlation study of drug sensitivity of PDO to the corresponding patient’s response to treatment. We have indicated it in the manuscript.

Comment 4: It would be good if in the "Conclusion" part the authors involve a brief comment on the ethical concerns of PDO research.

Answer: We have added a comment on the ethical concerns of PDO research in the Conclusion.

Reviewer 2 Report

The article presents a topical issue regarding personalized medicine. Neoadjuvant therapy in multiple neoplastic pathologies has proven effective in a high percentage of cases. The problem that arises in these therapies are those in small percentage who do not respond to neoadjuvant therapy and the loss of a crucial time (eg radiotherapy for rectal cancer that lasts 6 weeks can be a waste of time if the tumor is not radiosensitive) can lead the patient to palliative care. although initially something could be done for the patient.

Author Response

Comment 1: The article presents a topical issue regarding personalized medicine. Neoadjuvant therapy in multiple neoplastic pathologies has proven effective in a high percentage of cases. The problem that arises in these therapies are those in small percentage who do not respond to neoadjuvant therapy and the loss of a crucial time (eg radiotherapy for rectal cancer that lasts 6 weeks can be a waste of time if the tumor is not radiosensitive) can lead the patient to palliative care. although initially something could be done for the patient.

Answer: We appreciate the reviewer’s comment. As we mentioned in the manuscript, reducing inappropriate neoadjuvant treatment is important and PDO will be useful in this regard.